# Early Experience with the Biotronik Protego ICD Lead

**DOI:** 10.3390/jcm11237070

**Published:** 2022-11-29

**Authors:** Thomas Seiler, Christian Grebmer, Gabriela Hilfiker, Richard Kobza, Benjamin Berte

**Affiliations:** Heart Centre Lucerne, Luzerner Kantonsspital, 6000 Luzern, Switzerland

**Keywords:** Protego ICD leads, ICD, lead failure

## Abstract

Background: In the last decade, newer generation ICD leads have been developed based on mechanistic insides of priorly failing leads. The aim of our study was to assess the long-term performance and mechanisms of failure of the 2013-introduced Biotronik Protego ICD lead in a real-world population. Methods: All patients, who underwent implantation of a Protego ICD lead at the Heart Centre Lucerne (Lucerne, Switzerland) between November 2013 and March 2017, were followed up with semi-annual device-controls. The primary endpoint was defined as lead failure, secondary endpoints compromised all-cause death, (in)appropriate shocks and the need for reintervention. Results: A total of 64 patients (mean age 66.7 ± 8.7 years, 30% female) underwent implantation of a Protego ICD lead: 78% for primary prevention, 53% had underlying ischemic heart disease, and 40.6% had a dilated cardiomyopathy (DCM). Mean left ventricular ejection fraction (LVEF) was 32.6 ± 10.5%. A total of 24 patients were treated with cardiac resynchronization therapy (CRT), and their baseline LVEF improved from 27.8 ± 7.3% before to 39.8 ± 12.5 after implantation (*p* < 0.001). Mean time to follow-up was 5.5 ± 0.9 years. Overall, 14 patients (26.6%) suffered from at least one episode of sustained ventricular tachycardia; in total 10 patients (15.6%) died. Two patients experienced lead failure due to lead fracture after 5.5 and 5.7 years, which was clinically apparent by an abrupt rise in lead impedance (>2000 Ω) and by repetitive inappropriate shocks, respectively. Conclusions: In this retrospective observational study, the calculated annual lead failure rate of the Biotronik Protego ICD lead was 0.59% per patient—thus, the durability and long-term performance seem to be promising.

## 1. Introduction

Implantable cardioverter-defibrillators (ICD) reduce mortality in patients at risk of malignant arrhythmias [1,2]. However, lead failure is a dreaded complication and limitation of ICD therapy, and is associated with inappropriate ICD shocks or ineffective ICD therapy [3]. Over the last years, many patients have been affected by reduced lead survival rates of different high voltage leads [4,5,6].

By understanding the mechanisms and clinical manifestations of lead failure, different algorithms to detect lead failure and prevent inappropriate shocks have been developed [7,8]. Further, identification of weak points has helped to improve lead design and performance.

In 2013, the Biotronik Protego ICD lead was introduced on the market with a new DF4-connector providing a more durable and stable lead-device connection. This should, in theory, avoid insulation abrasions and reduce non-physiological high-rate sensing and improve lead performance.

To date, no systematic report on the long-term performance of Biotronik Protego ICD leads exists. For this purpose, we conducted a retrospective analysis of all patients who underwent implantation of a Protego ICD lead at the Heart Centre Lucerne between November 2013 and March 2017. The aim of our study was to assess lead failure rates and potential mechanisms of failure in a real-world population.

## 2. Methods

### 2.1. Population and Study Design

This study stems from the ongoing Lucerne Lead Registry, where all patients undergoing ICD implantation at the Heart Centre Lucerne (Lucerne, Switzerland) are registered. All patients, who underwent implantation of a Protego ICD lead at the Heart Centre Lucerne (Lucerne, Switzerland) between November 2013 and March 2017, were included for analysis, irrespective of their underlying heart disease or ICD indication. The study protocol was approved by our local institutional review board and all patients provided their written informed consent.

As a standard of care at our centre, all patients with an ICD undergo routine device control one day and six weeks after implantation and are subsequently controlled on a semi-annual basis.

### 2.2. The Protego Lead Family

The Biotronik Protego ICD leads represent the successors of the Biotronik Linox family and were first launched on the market in 2013. The 7.8F single- or dual-coil, tri- or quadripolar leads with active or passive fixation, have a silicone-based Silglide^®^ surface and a fractal, steroid-eluting coating which facilitates implantation, reduces friction and ensures low thresholds and optimal sensing. The Protek shock-coil design reduces tissue ingrowth and guarantees efficient shock delivery. The Biotronik tachy-lead history and lead characteristics are displayed in Figure 1.

The new Biotronik “DF4” lead connector is built with more inert materials (platinum iridium), which prevents the formation of metallic oxide layers that can potentially increase contact resistance leading to long-term sensing failure. Stabilizing the lead-device connection with more resistant materials reduces insulation defects after reinsertion of leads following device-replacements.

### 2.3. Study Endpoints

The primary endpoint was defined by lead failure, which was confirmed by the presence of any of the following criteria:Non-physiological high-rate signals, not assignable to electromagnetic interferences, myopotential or T wave oversensing;Sudden increase in right ventricular threshold or decrease of R wave sensing, without alternative explanation;Visual or fluoroscopic observation of an exposed or externalized conductor;Sudden change of long-term pace/sense or high voltage impedance (>100% increase or >50% decrease) or values outside the interval of 200–2000 Ω or 20–200 Ω, respectively.

Revisions due to lead dislodgements or perforations were not considered as failures, as they are mostly related to poor implantation quality and not directly related to longevity or quality of ICD leads.

The secondary endpoints were defined as all cause death, appropriate and inappropriate shocks, and the need for reinterventions. Further, therapy success of cardiac resynchronization therapy (CRT) was documented by evaluating the time course of left ventricular ejection fraction (LVEF), QRS morphology and width, as well as clinical symptoms following CRT implantation. Further, we evaluated possible mechanisms of CRT therapy failure.

### 2.4. Statistical Analysis

Continuous data are presented as mean ± standard deviation (SD); their normality distribution was confirmed with a Shapiro–Wilk test. Categorical variables are displayed as numbers (percentage). Statistical comparison between independent groups was performed with the Student’s t-test for parametric distributed data, and with the Mann–Whitney U test for non-parametric distributed data. For the outcome analysis, Cox regression models were calculated. Time-to-event after first implantation was estimated by applying the Kaplan–Meier method. A *p*-value < 0.05 was considered statistically significant. The statistical analyses were conducted with the STATA 17 software package (StataCorp LCC, Lakeway Drive, TX, USA).

## 3. Results

From November 2013 to March 2017, 64 Protego leads were implanted at our centre. A total of 54 procedures were performed as first implantations, and 10 procedures as revisions, where four patients had previously experienced inappropriate shocks.

Mean age at implantation was 66.7 ± 8.7 years, and a minority were female (30%). A total of 34 patients (53%) had underlying ischemic heart disease and 26 patients (40.6%) a dilated cardiomyopathy (DCM). A minority suffered from Brugada syndrome (*n* = 2), or hypertrophic cardiomyopathy (*n* = 1), and one patient survived sudden cardiac death due to myocarditis. LVEF at implantation was 32.6 ± 10.5%.

ICD was implanted for secondary prevention in 14 patients (22%) and for primary prevention in 50 patients (78%). Leads were, in general, implanted with one puncture site via the lateral subclavian vein. Baseline characteristics are displayed in Table 1.

In total, 24 patients were treated with cardiac resynchronisation therapy. Their baseline LVEF improved from 27.8 ± 7.3% before to 39.8 ± 12.5 (*p* < 0.001) after implantation. 3 patients had intermittent loss of biventricular pacing: one due to paroxysmal atrial fibrillation, one due to premature ventricular beats, and one because of programmed AV-hysteresis allowing intrinsic AV-conduction if present. Further details on CRT therapy are presented in Table 2.

## 4. Follow-Up

Detailed information regarding follow-up is displayed in Table 3. Mean time to follow-up was 5.5 ± 0.9 years. Overall, 14 patients (26.6%) suffered from at least one episode of sustained ventricular tachycardia, which were successfully terminated with antitachycardia pacing (ATP) or a shock delivery. In total 10 patients (15.6%) died; the corresponding time-to-event curve is shown in Figure 2.

Overall, three patients needed reintervention: one patient had right ventricular lead dislocation 2.7 months after implantation and two patients experienced lead failure due to lead fracture after 5.5 and 5.7 years, respectively. Lead failure was clinically apparent by an abrupt rise in lead impedance (>2000 Ω) in one patient, and by repetitive inappropriate shocks (*n* = 39) in the other patient.

## 5. Discussion

This study represents the first report on long-term performance and failure rate of the Biotronik Protego ICD lead in a real-world population. In our cohort (*n* = 64), two patients experienced lead fracture after 5.5 and 5.7 years, respectively, which corresponds to a calculated lead failure rate of 0.59% per patient per year. In one case, lead fracture was detected by an abrupt rise in lead impedance, whereas the other patient suffered a total of 39 inappropriate shocks.

In the last decade, some high-voltage ICD leads have been overshadowed by poor long-term performance and higher lead failure rates compared to other contemporary high voltage leads.

Lead survival rates after 5 years were previously estimated to be between 86 and 96% for the Biotronik Linox leads (Biotronik SE & Co. KG, Berlin, Germany), 83.6 and 90.1% for Medtronic Sprint Fidelis leads (Medtronic, Minneapolis, MN, USA) and 76 and 94.5% for St. Jude Riata leads (St. Jude Medical, St. Paul, MN, USA) [4,9,10]. Corresponding annual failure rates per patient were estimated to be at 2.9% for Linox leads, 1.17–2.6% for Riata leads and 2.23–4.8% for Sprint Fidelis leads [4,11,12]. These unacceptably high failure rates resulted in a systematic recall of the Riata and Sprint Fidelis leads in 2014, and highlighted the importance of continuous evaluation of lead performance in real-world populations—in particular since early results of the manufacturer’s approval studies reported highly reliable lead performance [11].

Thanks to numerous meticulous analyses, failure mechanisms and their clinical presentation could be identified. The failure of Riata leads was most commonly due to conductor externalization—a completely new mechanism, where damaged lead integrity is mainly coincidentally detected during thoracic fluoroscopy or device replacement [4]. In contrast, Fidelis and Linox leads are more prone to non-physiological high-rate signals, where patients present with inappropriate shocks [4]. Mechanistically, lead stress due to fraction and movement, as well as abrasion of silicon between the lead and generator, has been speculated to be majorly responsible for lead failure of Linox leads [5]. To counteract this mechanism, a more stable lead-to-can connection (DF-4 connector) has been developed in the newer generation Biotronik ICD leads, such as the Protego lead investigated here.

Indeed, the calculated annual failure rate of 0.59% per patient in our cohort, is comparable to failure rates of standard ICD leads (0.29–0.45% per person-year) and thus significantly better than the preceding Linox models [5,11]. Furthermore, both lead failures in our cohort were driven by lead fractures, and not primarily by an insulation abrasion or conductor externalization which was the main mechanism of failure of the Linox leads. Therefore, the performance of the Protego ICD leads seems to be promising.

In addition to technological advancements of lead designs, a deeper understanding of clinical and electrical manifestations of lead failure allowed the development of algorithms for detecting lead failure. For instance, the Abbott Secure Sense algorithm reliably inhibits inappropriate shocks by detecting right ventricular lead noise signals in case of discrepant electrocardiograms (ECG) of a near-field channel (derived from RV tip-to-ring or tip-to-coil) and a far-field channel (derived from RV tip-to-can or coil-to-can) [8,13,14]. The Medtronic lead integrity alert incorporates lead impedance trends and short R-R interval counters to detect potential lead failure, and automatically adapts device settings to discriminate ventricular arrhythmias [15]. This is extremely important, as lead failure can potentially hamper antitachycardia therapy, or evoke inappropriate shocks. Inappropriate shocks not only increase morbidity and mortality, but may also lead to psychological stress, as a lot of patients are heavily traumatised after perceiving them in full consciousness [16,17]. As these algorithms also have their limitations [18], it remains crucial to monitor patients for potential lead failure and to understand the underlying mechanism of failure.

Finally, as a lot of patients with an ICD indication also benefit from biventricular pacing, concomitant CRT with pacing indication is common. Lead failure in this population puts them at additional risk for therapy failure, as oversensing of non-physiological signals, higher pacing thresholds, or even loss of capture, disables biventricular pacing and can impair heart failure symptoms and prognosis.

### Limitations

The retrospective, single-centre design and the relatively small sample size (*n* = 64) clearly limit the generalisability and informative power of our study. As we also implanted other ICD leads during the study period, we cannot exclude a selection bias with lead selection. Further, explanted leads were not systematically analysed by the manufacturer to identify failure mechanisms.

Despite these limiting factors, the relatively long and regular follow-up of 5.5 ± 0.86 years should allow us to detect early lead failure in this real-world population.

## 6. Conclusions and Outlook

In this retrospective observational study, the calculated annual lead failure rate of the Biotronik Protego ICD lead was 0.59% per patient. The durability and long-term performance seem to be promising, but these need to be confirmed by larger upcoming multicentre studies that are currently underway (Protego DF4 Post-Approval Registry, NCT02243696). As lead failure remains the main limitation of ICD therapy and may impair successful CRT, systematic analysis of failure mechanisms is crucial to improve future lead designs and reduce lead failure rates.

## Figures and Tables

**Figure 1 jcm-11-07070-f001:**
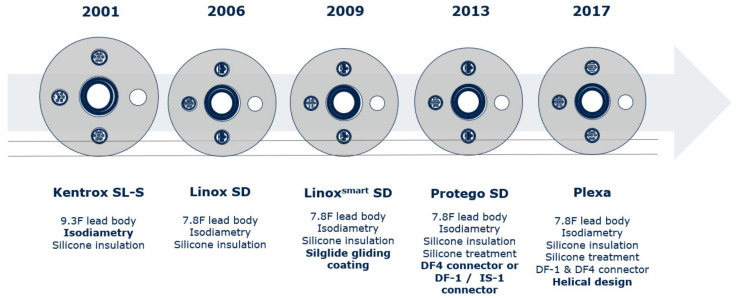
Biotronik ICD lead history (with friendly permission of BIOTRONIK).

**Figure 2 jcm-11-07070-f002:**
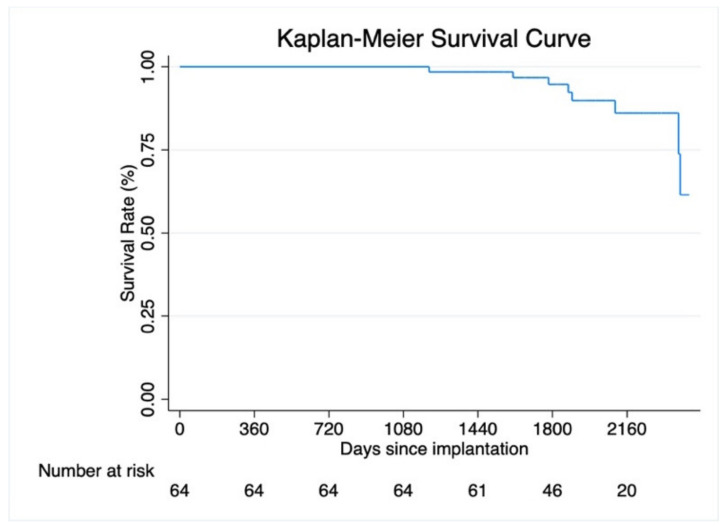
Time to event curve for all-cause death.

**Table 1 jcm-11-07070-t001:** Baseline characteristics.

	*n* = 64
Age (years)	66.7 ± 8.7
Female	19 (29.7)
BMI (kg/m^2^)	27.4 ± 5.8
LVEF (%)	32.6 ± 10.5
History of inappropriate shocks	4 (6.3)
Underlying heart disease	
Ischaemic	34 (53.1)
DCM	26 (40.6)
Other *	4 (6.3)
Indication for ICD	
Primary prevention	50 (78.2)
Secondary prevention	14 (21.8)
CRT	24 (37.5)
Implanted ICD system	
Single Chamber	28 (43.8)
Dual Chamber	12 (18.8)
Biventricular	24 (37.5)

BMI = Body mass index, LVEF = left ventricular ejection fraction, DCM = dilated cardiomyopathy, CRT = cardiac resynchronization therapy, ICD = intracardiac cardioverter defibrillator. * 2 patients with Brugada syndrome, 1 patient with hypertrophic cardiomyopathy and 1 patient post myocarditis.

**Table 2 jcm-11-07070-t002:** Technical details about cardiac resynchronisation therapy.

	*n* = 24		
CS position			
Anterior	2 (7.7)		
Anterolateral	5 (19)		
Posterior	4 (15)		
Posterolateral	6 (23)		
Lateral	7 (27)		
R in V1	5 (19)		
Q in aVL	3 (12)		
LBBB	17 (65)		
pLBBB	7 (27)		
	Before CRT	After CRT	*p*-value
LVEF	27.8 ± 7.3	39.8 ± 12.5	*p* < 0.001
PQ	170 ± 38	147 ± 29	*p* = 0.06
QRS	151 ± 22	145 ± 22	*p* = 0.40

CS = coronary sinus, LVEF = left ventricular ejection fraction, PQ = PQ-time, QRS = QRS duration, LBBB = left bundle branch block, pLBBB = partial left bundle branch block, CRT = cardiac resynchronisation therapy.

**Table 3 jcm-11-07070-t003:** Follow-up.

	*n* = 64
Time to follow-up (months)	65.8 ± 10.3
Reintervention *	3 (4.7)
Lead failure	2 (3)
Inappropriate shock	1 (1.6)
ATP/appropriate shock	14 (22)
All-cause death	10 (15.6)

ATP = antitachycardia pacing. * One reintervention was performed due to right ventricular lead dislocation 2.7 months after implantation; two reinterventions were performed due to lead fracture after 5.5 and 5.7 years, respectively.

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
