# Peer review of "Early Experience with the Biotronik Protego ICD Lead"

_jcm, 2022, doi:10.3390/jcm11237070_

Round 1
Reviewer 1 Report
The article by Thomas Seiler et al. entitled “Early experience with the Biotronik Protego ICD lead” aimed to assess the long-term performance and mechanisms of failure of the Biotronik Protego ICD lead in a real-world population. This is the retrospective, single-center observational study with the small sample size but relatively long follow-up. The results of the study are of major importance, because the lead failure presents the weakest point of this lifesaving therapy. While the topic could be of interest, the necessary corrections must be made. I have listed some points for improvement below.
Main concern:
1. Inappropriate statistical methods are used in the study.
2. There are wrong values in the results section. Table 1: CRT was implanted in 26, but biventricular ICD system was implanted in 24. The total number of “indications for implantation” is 66 (26+14+26). The results must be checked and corrected. The same in Table 2 and throughout the text.
3. The quality of Figure 2 is very low. Moreover, I think both Figure 1 and Figure 2 are unnecessary in the manuscript and should be removed. Lines 64-65 and 74-75 should also be removed, consequently.
4. Line 136: “at least form one” I suppose it should be “from”.
5. Lines 138-139 and Figure 3: time-to-event curve (Kaplan-Meier) has no relation to the aim of the study.
6. Ref 18 is incomplete.
I suggest major revisions and additional review for the manuscript and look forward to a revised version.
Author Response
Reply to the Editors and Reviewers
The authors wish to express their thanks to the editors and three reviewers involved in the process. Their interest and excellent remarks clearly helped to improve the manuscript. Below are all our comments to the individual questions raised. Changes to the manuscript are specifically indicated. Reviewer comments are in italics followed by our response.
Reviewer 1
- Comment 1: Inappropriate statistical methods are used in the study.
Answer: We apologize for our rather generalized and not clearly specified “statistics section”. All presented continuous data were confirmed to be parametric distributed with the Shapiro-Wilk-Test and are consequently presented as mean ± standard deviation. Statistical comparison between groups were performed with a student’s t-test (for parametric distributed data) or Man-Whitney-U test (for non-parametric distributed data).
We adapted the statistics section as follows:
Page 5, lines 1--9
“Continuous data are presented as mean ± standard deviation (SD) - their normality distribution was confirmed with a Shapiro-Wilk-Test. Categorical variables are displayed as numbers (percentage). Statistical comparison between independent groups was performed with the students t-test for parametric distributed data and with the Man-Whitney-U test for non-parametric distributed data. For the outcome analysis, Cox regression models were calculated. Time-to-event after first implantation was estimated by applying the Kaplan-Meier method. A p-value <0.05 was considered statistically significant. The statistical analyses were conducted with the STATA 17 software package (StataCorp LCC, Lakeway Drive, Texas, USA).”
- Comment 2: There are wrong values in the results section. Table 1: CRT was implanted in 26, but biventricular ICD system was implanted in 24. The total number of “indications for implantation” is 66 (24 + 14 + 26). These results must be checked and corrected. The same in Table 2 and throughout the text.
Answer: Thank you for pointing out this problem. In total, 14 patients underwent ICD implantation for secondary prevention and 50 patients for primary prevention. Overall, 24 patients (out of 64) had an additional indication for CRT.
We apologize for this error and changed the results section and tables as followed:
Page 6, lines 10-13
“ICD was implanted for secondary prevention in 14 patients (22%) and for primary prevention in 50 patients (78%). […] 24 patients were treated with cardiac resynchronization therapy.”
Table 1: Baseline-characteristics
|
|
n=64 |
|
Age (years) |
66.7 ± 8.7 |
|
Female |
19 (29.7) |
|
BMI (kg/m2) |
27.4 ± 5.8 |
|
LVEF (%) |
32.6 ± 10.5 |
|
History of inappropriate shocks |
4 (6.3) |
|
Underlying heart disease |
|
|
Ischemic |
34 (53.1) |
|
DCM |
26 (40.6) |
|
Other* |
4 (6.25) |
|
Indication for ICD |
|
|
Primary prevention |
50 (78.2) |
|
Secondary prevention |
14 (21.8) |
|
CRT |
24 (37.5) |
|
Implanted ICD system |
|
|
Single Chamber |
28 (43.8) |
|
Dual Chamber |
12 (18.8) |
|
Biventricular |
24 (37.5) |
BMI = Body mass index, LVEF = left ventricular ejection fraction, DCM = dilated cardiomyopathy, CRT = cardiac resynchronization therapy, ICD = intracardiac cardioverter defibrillator
* 2 patients with Brugada syndrome, 1 patient with hypertrophic cardiomyopathy and 1 patient post myocarditis
- Comment 3: The quality of Figure 2 is very low. Moreover, I think both Figures 1 and 2 are unnecessary in the manuscript and should be removed. Lines 64-65 and 74-75 should also be removed, consequently.
Answer: We removed figure 2 and the corresponding lines in the manuscript. Figure 1 may be of interest to some readers, who are not familiar with the Biotronik ICD leads, as it provides some interesting background information. We therefore included Figure 1 in the supplements and changed lines 74-75 in the manuscript as follows:
Page 4, lines 7-8
“ The Biotronik tachy-lead history and lead characteristics are displayed in supplementary figure 1.”
- Comment 4: Line 136 “at least form on” I suppose it should be “from”
Answer: Thank you for this hint, we changed the sentence according to your suggestion as follows:
Page 13, lines 13-14
“14 patients (26.6%) suffered at least from one episode of sustained ventricular tachycardia, which were successfully terminated with antitachycardia pacing (ATP) or a shock delivery. “
- Comment 5: Lines 138-139 and Figure 3: time-to-event curve (Kaplan-Meier) has no relation to the aim of the study.
Answer: We agree that the time-to-event curve (Figure 3) is not of main interest for our study. However, death rate is in general an interesting information and a marker of (preexisting) morbidity. Further, failure to provide ICD therapy may ultimately lead to death. Also, inappropriate shocks have been associated with increased mortality. Based on the above reasoning, we have left figure 3 in the manuscript, but reduced in size as suggested by Reviewer 2.
As figure 1 is now in the supplements and figure 2 was removed from the script, figure 3 is renamed to “figure 1”.
- Comment 6: Ref 18 is incomplete.
Answer: We apologize for the missing information and corrected reference 18 as follows:
Page 13, lines 44-46
“Seiler, T., et al., Recurrent implantable cardioverter-defibrillator shocks due to automatic deactivation of a right ventricular lead noise discrimination algorithm. HeartRhythm Case Rep, 2022. 8(10): p. 695-698.”
Reviewer 2
- Comment 1: Figure 2: An entire page (out of 8) is reserved for figure 2, which is reproduced from another publication/technical manual. Which is the purpose of figure 2 with respect to the goal of the paper? In addition, Figure 2 is blurred, and its quality is very low.
Answer: Thank you for this very important question and for pointing out the poor image quality of Figure 2. We completely agree that figure 2 is not directly related to the goal of the paper. Our idea was to provide technical insights, which may explain the improved lead performance compared to the preceding “Linox” models. As proposed by Reviewer 1, we removed figure 2 and the indicative sentence from our manuscript.
- Comment 2: Page 4, line 90: The authors write: Revisions due to lead dislodgements or perforations were not considered as failures. Why was this decision made? How is it motivated?
Answer: The principal goal of our study, was to assess the long-term performance of the Protego lead. Dislodgments and perforations are usually an early problem after implantation and more related to poor implantation quality, rather than poor lead performance. Because of this, we did not include these events as markers of lead quality or longevity.
However, in our cohort only one patient suffered lead dislodgment, which we highlighted in the result section as follows:
Page 6, lines 25-27
“Overall, 3 patients needed reintervention: one patient had right ventricular lead dislocation 2.7 months after implantation and two patients experienced lead failure due to lead fracture after 5.5 and 5.7 years, respectively. “
- Comment 3: Page 4, Line 91: The authors write: The secondary endpoints were defined as all cause death, appropriate and inappropriate shocks and the need for reinterventions. Why are these 3 very different endpoints grouped together? A clear motivation for this choice is needed.
Answer: We need to clarify, that these secondary endpoints are not grouped to a composite endpoint, but are rather individual secondary sub-items, which are of clinical interest: As mentioned in “comment 5, reviewer 1” death is an important marker of (preexisting) morbidity and may also be indirectly influenced by lead failure in case of ICD therapy failure. Appropriate and inappropriate shocks are of big interest for our patients, as they are both associated with increased mortality. Finally, the need for reintervention also takes into account lead dislodgements and perforations, which were not included in the primary endpoint.
- Comment 4: Statistical analysis section: This section is very unclear. The authors mention a long list of tests without providing information on when a test is used and why. Only the relevant test/measures should be listed, and it should be clearly stated on which data they are used.
Answer: We apologize for our rather generalized and not clearly specified “statistics section”. All presented continuous data were confirmed to be parametrically distributed with the Shapiro-Wilk-Test and are consequently presented as mean ± standard deviation. Statistical comparison between groups were performed with a student’s t-test or Man-Whitney-U test, depending on their parametric or non-parametric distribution (e.g. categorical data).
We adapted the statistics section as follows:
Page 5, lines 1--9
“Continuous data are presented as mean ± standard deviation (SD) - their normality distribution was confirmed with a Shapiro-Wilk-Test. Categorical variables are displayed as numbers (percentage). Statistical comparison between independent groups was performed with the students t-test for parametric distributed data and with the Man-Whitney-U test for non-parametric distributed data. For the outcome analysis, Cox regression models were calculated. Time-to-event after first implantation was estimated by applying the Kaplan-Meier method. A p-value <0.05 was considered statistically significant. The statistical analyses were conducted with the STATA 17 software package (StataCorp LCC, Lakeway Drive, Texas, USA).”
- Comment 5: Additionally, the authors write: “Normality distribution was tested with visual inspection of the skewness and kurtosis as well as using a Shakiro-Wilk-Test. Why was visual inspection used? Why sometimes visual inspection and sometimes other tests? On which data were these tests used?
Answer: We apologize once more for our rather unspecified and generally written statistics section. In detail, we tested all our continuous data for normality distribution with the Shakiro-Wilk-Test. Visual inspection was not used for our data.
The statistics section was adapted as mentioned above.
- Comment 6: Statistical analysis section: The authors also write: For statistical comparison between groups the Man-Whitney-U test, chi-square test, Fishers exact test or students t-test were used as appropriate. When are these tests used? On which data is every test used?
Answer: As we deal with independent data (and groups), the students t-test was applied for parametric data and the Man-Whitney-U test for non-parametric data. We apologize for this mistake and changed our statistics section as written in “comment 4”.
- Comment 7: Page 5: What do the authors mean by “of programmed AV-hysteresis”? A clarification is needed about this statement.
Answer: In the up-mentioned sentence, we try to explain, why patients had an insufficiently low rate of biventricular pacing.
In patients without complete AV-block, a programmed AV-delay shorter than the intrinsic AV conduction is key, to achieve a sufficient biventricular pacing. If the programmed AV-delay is longer than the intrinsic AV-conduction, biventricular pacing will be lost. In one patient with loss of biventricular pacing, an AV-hysteresis was programmed. The AV-hysteresis algorithm intermittently prolongs the AV-delay to allow intrinsic AV conduction if present (which was the case in our patient).
We changed the sentence as follows:
Page 6, line 14-17
“3 patients had intermittent loss of biventricular pacing: one due to paroxysmal atrial fibrillation, one due to premature ventricular beats and one because of programmed AV-hysteresis allowing intrinsic AV-conduction if present.”
- Comment 8: Table 2: The relevance of Table 2 with respect to the goal of the paper is unclear. How are the details related to resynchronization therapy related to the durability and long-term performance of the Biotronik Protego ICD leads?
Answer: We agree that details of resynchronization therapy are not directly related to the durability of Biotronik Protego ICD leads. Information in table 2 is not directly related to the principal study, but may be of interest for a better characterization of our cohort.
Vice versa, RV lead failure with oversensing may interact with CRT therapy and reduce its efficacy, which may impact the heart failure burden and clinical prognosis. We therefore think, that characterizing the CRT cohort in our study is of interest.
Comment 9: Figure 3: What is the purpose of this very large figure? The goal of the paper is to provide information about the failure and mode of failure of the leads: how is the time to all cause of death related to it? This figure should be removed from the paper or explained and reduced in size.
Answer: Thank you for this important comment. We agree with you, that mortality has no impact on the longevity of the ICD leads. But, as lead failure may impact death by hindering appropriate shocks or by provoking inappropriate shocks, it may be of interest to specify all cause death in this cohort. Since mortality is usually higher in multimorbid patients, all death may also be an indirect parameter of morbidity.
Based on the above reasoning, we have left Figure 3 in the manuscript, but reduced in size as suggested by Reviewer 2.
Figure 3 was renamed to Figure 1.
Reviewer 3
We would like to thank to Reviewer 3 for the constructive criticism, and well summarized feedback.
(There are no specific comments to answer).
Reviewer 2 Report
The goal of this paper is to identify potential mechanisms of failure for the Biotronik Protego ICD lead in a real-world population. The study considers 64 patients in a retrospective study. The presented results are interesting and useful for researchers and clinicians in the field, but I have concerns regarding the presented methodology and data presentation. In addition, in an already short manuscript, significant space is taken by figures which are not relevant to the goal of the paper.
The following points are meant to provide some suggestions/comments to the authors:
1. Figure 2: An entire page (out of 8) is reserved for figure 2, which is reproduced from another publication/technical manual. What is the purpose of figure 2 with respect to the goal of the paper? In addition, Figure 2 is blurred, and its quality is very low.
2. Page 4, line 90: The authors write: “Revisions due to lead dislodgements or perforations were not considered as failures.” Why was this decision made? How is it motivated?
3. Page 4, line 91: The authors write: “The secondary endpoints were defined as all cause death, appropriate and inappropriate shocks and the need for reinterventions.” Why are these 3 very different endpoints grouped together? A clear motivation for this choice is needed.
4. Statistical analysis section: This section is very unclear. The authors mention a long list of tests without providing information on when a test is used and why. Only the relevant tests/measures should be listed and it should be clearly stated on which data they are used. For example, the authors write: “Continuous data are presented as mean ± standard deviation (SD) or median (interquartile range (IQR)) as appropriate.” Why do they adopt mean and median interchangeably? Which data are represented by mean ± SD and which data are represented by median and IQR?
5. Statistical analysis section: Additionally, the authors write: “Normality distribution was tested with visual inspection of the skewness and kurtosis as well as using a Shapiro-Wilk-Test.” Why was visual inspection used? Why sometimes visual inspection and sometimes other tests? On which data were these tests used?
6. Statistical analysis section: The authors also write: “For statistical comparison between groups the Man-Whitney-U test, chi-square test, Fishers exact test or students t-test were used as appropriate.” When are these tests used? On which data is every test used?
7. Page 5: What do the authors mean by “of programmed AV-hysteresis”? A clarification is needed about this statement.
8. Table 2: The relevance of Table 2 with respect to the goal of the paper is unclear. How are the details related to resynchronization therapy related to the durability and long-term performance of the Biotronik Protego ICD leads?
9. Figure 3: What is the purpose of this very large figure? The goal of the paper is to provide information about the failure and mode of failure of the leads: how is the time to all cause of death related to it? This figure should be removed from the paper or explained and reduced in size.
Author Response
The authors wish to express their thanks to the editors and three reviewers involved in the process. Their interest and excellent remarks clearly helped to improve the manuscript. Below are all our comments to the individual questions raised. Changes to the manuscript are specifically indicated. Reviewer comments are in italics followed by our response.
Reviewer 2
- Comment 1: Figure 2: An entire page (out of 8) is reserved for figure 2, which is reproduced from another publication/technical manual. Which is the purpose of figure 2 with respect to the goal of the paper? In addition, Figure 2 is blurred, and its quality is very low.
Answer: Thank you for this very important question and for pointing out the poor image quality of Figure 2. We completely agree that figure 2 is not directly related to the goal of the paper. Our idea was to provide technical insights, which may explain the improved lead performance compared to the preceding “Linox” models. As proposed by Reviewer 1, we removed figure 2 and the indicative sentence from our manuscript.
- Comment 2: Page 4, line 90: The authors write: Revisions due to lead dislodgements or perforations were not considered as failures. Why was this decision made? How is it motivated?
Answer: The principal goal of our study, was to assess the long-term performance of the Protego lead. Dislodgments and perforations are usually an early problem after implantation and more related to poor implantation quality, rather than poor lead performance. Because of this, we did not include these events as markers of lead quality or longevity.
However, in our cohort only one patient suffered lead dislodgment, which we highlighted in the result section as follows:
Page 6, lines 25-27
“Overall, 3 patients needed reintervention: one patient had right ventricular lead dislocation 2.7 months after implantation and two patients experienced lead failure due to lead fracture after 5.5 and 5.7 years, respectively. “
- Comment 3: Page 4, Line 91: The authors write: The secondary endpoints were defined as all cause death, appropriate and inappropriate shocks and the need for reinterventions. Why are these 3 very different endpoints grouped together? A clear motivation for this choice is needed.
Answer: We need to clarify, that these secondary endpoints are not grouped to a composite endpoint, but are rather individual secondary sub-items, which are of clinical interest: As mentioned in “comment 5, reviewer 1” death is an important marker of (preexisting) morbidity and may also be indirectly influenced by lead failure in case of ICD therapy failure. Appropriate and inappropriate shocks are of big interest for our patients, as they are both associated with increased mortality. Finally, the need for reintervention also takes into account lead dislodgements and perforations, which were not included in the primary endpoint.
- Comment 4: Statistical analysis section: This section is very unclear. The authors mention a long list of tests without providing information on when a test is used and why. Only the relevant test/measures should be listed, and it should be clearly stated on which data they are used.
Answer: We apologize for our rather generalized and not clearly specified “statistics section”. All presented continuous data were confirmed to be parametrically distributed with the Shapiro-Wilk-Test and are consequently presented as mean ± standard deviation. Statistical comparison between groups were performed with a student’s t-test or Man-Whitney-U test, depending on their parametric or non-parametric distribution (e.g. categorical data).
We adapted the statistics section as follows:
Page 5, lines 1--9
“Continuous data are presented as mean ± standard deviation (SD) - their normality distribution was confirmed with a Shapiro-Wilk-Test. Categorical variables are displayed as numbers (percentage). Statistical comparison between independent groups was performed with the students t-test for parametric distributed data and with the Man-Whitney-U test for non-parametric distributed data. For the outcome analysis, Cox regression models were calculated. Time-to-event after first implantation was estimated by applying the Kaplan-Meier method. A p-value <0.05 was considered statistically significant. The statistical analyses were conducted with the STATA 17 software package (StataCorp LCC, Lakeway Drive, Texas, USA).”
- Comment 5: Additionally, the authors write: “Normality distribution was tested with visual inspection of the skewness and kurtosis as well as using a Shakiro-Wilk-Test. Why was visual inspection used? Why sometimes visual inspection and sometimes other tests? On which data were these tests used?
Answer: We apologize once more for our rather unspecified and generally written statistics section. In detail, we tested all our continuous data for normality distribution with the Shakiro-Wilk-Test. Visual inspection was not used for our data.
The statistics section was adapted as mentioned above.
- Comment 6: Statistical analysis section: The authors also write: For statistical comparison between groups the Man-Whitney-U test, chi-square test, Fishers exact test or students t-test were used as appropriate. When are these tests used? On which data is every test used?
Answer: As we deal with independent data (and groups), the students t-test was applied for parametric data and the Man-Whitney-U test for non-parametric data. We apologize for this mistake and changed our statistics section as written in “comment 4”.
- Comment 7: Page 5: What do the authors mean by “of programmed AV-hysteresis”? A clarification is needed about this statement.
Answer: In the up-mentioned sentence, we try to explain, why patients had an insufficiently low rate of biventricular pacing.
In patients without complete AV-block, a programmed AV-delay shorter than the intrinsic AV conduction is key, to achieve a sufficient biventricular pacing. If the programmed AV-delay is longer than the intrinsic AV-conduction, biventricular pacing will be lost. In one patient with loss of biventricular pacing, an AV-hysteresis was programmed. The AV-hysteresis algorithm intermittently prolongs the AV-delay to allow intrinsic AV conduction if present (which was the case in our patient).
We changed the sentence as follows:
Page 6, line 14-17
“3 patients had intermittent loss of biventricular pacing: one due to paroxysmal atrial fibrillation, one due to premature ventricular beats and one because of programmed AV-hysteresis allowing intrinsic AV-conduction if present.”
- Comment 8: Table 2: The relevance of Table 2 with respect to the goal of the paper is unclear. How are the details related to resynchronization therapy related to the durability and long-term performance of the Biotronik Protego ICD leads?
Answer: We agree that details of resynchronization therapy are not directly related to the durability of Biotronik Protego ICD leads. Information in table 2 is not directly related to the principal study, but may be of interest for a better characterization of our cohort.
Vice versa, RV lead failure with oversensing may interact with CRT therapy and reduce its efficacy, which may impact the heart failure burden and clinical prognosis. We therefore think, that characterizing the CRT cohort in our study is of interest.
Comment 9: Figure 3: What is the purpose of this very large figure? The goal of the paper is to provide information about the failure and mode of failure of the leads: how is the time to all cause of death related to it? This figure should be removed from the paper or explained and reduced in size.
Answer: Thank you for this important comment. We agree with you, that mortality has no impact on the longevity of the ICD leads. But, as lead failure may impact death by hindering appropriate shocks or by provoking inappropriate shocks, it may be of interest to specify all cause death in this cohort. Since mortality is usually higher in multimorbid patients, all death may also be an indirect parameter of morbidity.
Based on the above reasoning, we have left Figure 3 in the manuscript, but reduced in size as suggested by Reviewer 2.
Figure 3 was renamed to Figure 1.
Reviewer 1
- Comment 1: Inappropriate statistical methods are used in the study.
Answer: We apologize for our rather generalized and not clearly specified “statistics section”. All presented continuous data were confirmed to be parametric distributed with the Shapiro-Wilk-Test and are consequently presented as mean ± standard deviation. Statistical comparison between groups were performed with a student’s t-test (for parametric distributed data) or Man-Whitney-U test (for non-parametric distributed data).
We adapted the statistics section as follows:
Page 5, lines 1--9
“Continuous data are presented as mean ± standard deviation (SD) - their normality distribution was confirmed with a Shapiro-Wilk-Test. Categorical variables are displayed as numbers (percentage). Statistical comparison between independent groups was performed with the students t-test for parametric distributed data and with the Man-Whitney-U test for non-parametric distributed data. For the outcome analysis, Cox regression models were calculated. Time-to-event after first implantation was estimated by applying the Kaplan-Meier method. A p-value <0.05 was considered statistically significant. The statistical analyses were conducted with the STATA 17 software package (StataCorp LCC, Lakeway Drive, Texas, USA).”
- Comment 2: There are wrong values in the results section. Table 1: CRT was implanted in 26, but biventricular ICD system was implanted in 24. The total number of “indications for implantation” is 66 (24 + 14 + 26). These results must be checked and corrected. The same in Table 2 and throughout the text.
Answer: Thank you for pointing out this problem. In total, 14 patients underwent ICD implantation for secondary prevention and 50 patients for primary prevention. Overall, 24 patients (out of 64) had an additional indication for CRT.
We apologize for this error and changed the results section and tables as followed:
Page 6, lines 10-13
“ICD was implanted for secondary prevention in 14 patients (22%) and for primary prevention in 50 patients (78%). […] 24 patients were treated with cardiac resynchronization therapy.”
Table 1: Baseline-characteristics
|
|
n=64 |
|
Age (years) |
66.7 ± 8.7 |
|
Female |
19 (29.7) |
|
BMI (kg/m2) |
27.4 ± 5.8 |
|
LVEF (%) |
32.6 ± 10.5 |
|
History of inappropriate shocks |
4 (6.3) |
|
Underlying heart disease |
|
|
Ischemic |
34 (53.1) |
|
DCM |
26 (40.6) |
|
Other* |
4 (6.25) |
|
Indication for ICD |
|
|
Primary prevention |
50 (78.2) |
|
Secondary prevention |
14 (21.8) |
|
CRT |
24 (37.5) |
|
Implanted ICD system |
|
|
Single Chamber |
28 (43.8) |
|
Dual Chamber |
12 (18.8) |
|
Biventricular |
24 (37.5) |
BMI = Body mass index, LVEF = left ventricular ejection fraction, DCM = dilated cardiomyopathy, CRT = cardiac resynchronization therapy, ICD = intracardiac cardioverter defibrillator
* 2 patients with Brugada syndrome, 1 patient with hypertrophic cardiomyopathy and 1 patient post myocarditis
- Comment 3: The quality of Figure 2 is very low. Moreover, I think both Figures 1 and 2 are unnecessary in the manuscript and should be removed. Lines 64-65 and 74-75 should also be removed, consequently.
Answer: We removed figure 2 and the corresponding lines in the manuscript. Figure 1 may be of interest to some readers, who are not familiar with the Biotronik ICD leads, as it provides some interesting background information. We therefore included Figure 1 in the supplements and changed lines 74-75 in the manuscript as follows:
Page 4, lines 7-8
“ The Biotronik tachy-lead history and lead characteristics are displayed in supplementary figure 1.”
- Comment 4: Line 136 “at least form on” I suppose it should be “from”
Answer: Thank you for this hint, we changed the sentence according to your suggestion as follows:
Page 13, lines 13-14
“14 patients (26.6%) suffered at least from one episode of sustained ventricular tachycardia, which were successfully terminated with antitachycardia pacing (ATP) or a shock delivery. “
- Comment 5: Lines 138-139 and Figure 3: time-to-event curve (Kaplan-Meier) has no relation to the aim of the study.
Answer: We agree that the time-to-event curve (Figure 3) is not of main interest for our study. However, death rate is in general an interesting information and a marker of (preexisting) morbidity. Further, failure to provide ICD therapy may ultimately lead to death. Also, inappropriate shocks have been associated with increased mortality. Based on the above reasoning, we have left figure 3 in the manuscript, but reduced in size as suggested by Reviewer 2.
As figure 1 is now in the supplements and figure 2 was removed from the script, figure 3 is renamed to “figure 1”.
- Comment 6: Ref 18 is incomplete.
Answer: We apologize for the missing information and corrected reference 18 as follows:
Page 13, lines 44-46
“Seiler, T., et al., Recurrent implantable cardioverter-defibrillator shocks due to automatic deactivation of a right ventricular lead noise discrimination algorithm. HeartRhythm Case Rep, 2022. 8(10): p. 695-698.”
Reviewer 3
We would like to thank to Reviewer 3 for the constructive criticism, and well summarized feedback.
(There are no specific comments to answer).
Reviewer 3 Report
Opinion for the article entitled "Early experience with the Biotronik Protego ICD lead".
This paper is a retrospective analysis of the long-term performance of the Biotronik Protego ICD electrodes. The authors investigated the potentially frightening problem of ICD failure rates and potential failure mechanisms in the real population. Both the research methodology and the presentation of the results do not raise any objections. The presented work is written concisely, the conclusions are not exaggerated, and the authors are aware of the limitations of retrospective single-center design and the relatively small sample size.
Despite these limitations, the work brings interesting and clinically useful knowledge in the field of Protego ICD biotechnology.
Author Response
The authors wish to express their thanks to the editors and three reviewers involved in the process. Their interest and excellent remarks clearly helped to improve the manuscript. Below are all our comments to the individual questions raised. Changes to the manuscript are specifically indicated. Reviewer comments are in italics followed by our response.
Reviewer 3
We would like to thank to Reviewer 3 for the constructive criticism, and well summarized feedback.
(There are no specific comments to answer).
Reviewer 2
- Comment 1: Figure 2: An entire page (out of 8) is reserved for figure 2, which is reproduced from another publication/technical manual. Which is the purpose of figure 2 with respect to the goal of the paper? In addition, Figure 2 is blurred, and its quality is very low.
Answer: Thank you for this very important question and for pointing out the poor image quality of Figure 2. We completely agree that figure 2 is not directly related to the goal of the paper. Our idea was to provide technical insights, which may explain the improved lead performance compared to the preceding “Linox” models. As proposed by Reviewer 1, we removed figure 2 and the indicative sentence from our manuscript.
- Comment 2: Page 4, line 90: The authors write: Revisions due to lead dislodgements or perforations were not considered as failures. Why was this decision made? How is it motivated?
Answer: The principal goal of our study, was to assess the long-term performance of the Protego lead. Dislodgments and perforations are usually an early problem after implantation and more related to poor implantation quality, rather than poor lead performance. Because of this, we did not include these events as markers of lead quality or longevity.
However, in our cohort only one patient suffered lead dislodgment, which we highlighted in the result section as follows:
Page 6, lines 25-27
“Overall, 3 patients needed reintervention: one patient had right ventricular lead dislocation 2.7 months after implantation and two patients experienced lead failure due to lead fracture after 5.5 and 5.7 years, respectively. “
- Comment 3: Page 4, Line 91: The authors write: The secondary endpoints were defined as all cause death, appropriate and inappropriate shocks and the need for reinterventions. Why are these 3 very different endpoints grouped together? A clear motivation for this choice is needed.
Answer: We need to clarify, that these secondary endpoints are not grouped to a composite endpoint, but are rather individual secondary sub-items, which are of clinical interest: As mentioned in “comment 5, reviewer 1” death is an important marker of (preexisting) morbidity and may also be indirectly influenced by lead failure in case of ICD therapy failure. Appropriate and inappropriate shocks are of big interest for our patients, as they are both associated with increased mortality. Finally, the need for reintervention also takes into account lead dislodgements and perforations, which were not included in the primary endpoint.
- Comment 4: Statistical analysis section: This section is very unclear. The authors mention a long list of tests without providing information on when a test is used and why. Only the relevant test/measures should be listed, and it should be clearly stated on which data they are used.
Answer: We apologize for our rather generalized and not clearly specified “statistics section”. All presented continuous data were confirmed to be parametrically distributed with the Shapiro-Wilk-Test and are consequently presented as mean ± standard deviation. Statistical comparison between groups were performed with a student’s t-test or Man-Whitney-U test, depending on their parametric or non-parametric distribution (e.g. categorical data).
We adapted the statistics section as follows:
Page 5, lines 1--9
“Continuous data are presented as mean ± standard deviation (SD) - their normality distribution was confirmed with a Shapiro-Wilk-Test. Categorical variables are displayed as numbers (percentage). Statistical comparison between independent groups was performed with the students t-test for parametric distributed data and with the Man-Whitney-U test for non-parametric distributed data. For the outcome analysis, Cox regression models were calculated. Time-to-event after first implantation was estimated by applying the Kaplan-Meier method. A p-value <0.05 was considered statistically significant. The statistical analyses were conducted with the STATA 17 software package (StataCorp LCC, Lakeway Drive, Texas, USA).”
- Comment 5: Additionally, the authors write: “Normality distribution was tested with visual inspection of the skewness and kurtosis as well as using a Shakiro-Wilk-Test. Why was visual inspection used? Why sometimes visual inspection and sometimes other tests? On which data were these tests used?
Answer: We apologize once more for our rather unspecified and generally written statistics section. In detail, we tested all our continuous data for normality distribution with the Shakiro-Wilk-Test. Visual inspection was not used for our data.
The statistics section was adapted as mentioned above.
- Comment 6: Statistical analysis section: The authors also write: For statistical comparison between groups the Man-Whitney-U test, chi-square test, Fishers exact test or students t-test were used as appropriate. When are these tests used? On which data is every test used?
Answer: As we deal with independent data (and groups), the students t-test was applied for parametric data and the Man-Whitney-U test for non-parametric data. We apologize for this mistake and changed our statistics section as written in “comment 4”.
- Comment 7: Page 5: What do the authors mean by “of programmed AV-hysteresis”? A clarification is needed about this statement.
Answer: In the up-mentioned sentence, we try to explain, why patients had an insufficiently low rate of biventricular pacing.
In patients without complete AV-block, a programmed AV-delay shorter than the intrinsic AV conduction is key, to achieve a sufficient biventricular pacing. If the programmed AV-delay is longer than the intrinsic AV-conduction, biventricular pacing will be lost. In one patient with loss of biventricular pacing, an AV-hysteresis was programmed. The AV-hysteresis algorithm intermittently prolongs the AV-delay to allow intrinsic AV conduction if present (which was the case in our patient).
We changed the sentence as follows:
Page 6, line 14-17
“3 patients had intermittent loss of biventricular pacing: one due to paroxysmal atrial fibrillation, one due to premature ventricular beats and one because of programmed AV-hysteresis allowing intrinsic AV-conduction if present.”
- Comment 8: Table 2: The relevance of Table 2 with respect to the goal of the paper is unclear. How are the details related to resynchronization therapy related to the durability and long-term performance of the Biotronik Protego ICD leads?
Answer: We agree that details of resynchronization therapy are not directly related to the durability of Biotronik Protego ICD leads. Information in table 2 is not directly related to the principal study, but may be of interest for a better characterization of our cohort.
Vice versa, RV lead failure with oversensing may interact with CRT therapy and reduce its efficacy, which may impact the heart failure burden and clinical prognosis. We therefore think, that characterizing the CRT cohort in our study is of interest.
Comment 9: Figure 3: What is the purpose of this very large figure? The goal of the paper is to provide information about the failure and mode of failure of the leads: how is the time to all cause of death related to it? This figure should be removed from the paper or explained and reduced in size.
Answer: Thank you for this important comment. We agree with you, that mortality has no impact on the longevity of the ICD leads. But, as lead failure may impact death by hindering appropriate shocks or by provoking inappropriate shocks, it may be of interest to specify all cause death in this cohort. Since mortality is usually higher in multimorbid patients, all death may also be an indirect parameter of morbidity.
Based on the above reasoning, we have left Figure 3 in the manuscript, but reduced in size as suggested by Reviewer 2.
Figure 3 was renamed to Figure 1.
Reviewer 1
- Comment 1: Inappropriate statistical methods are used in the study.
Answer: We apologize for our rather generalized and not clearly specified “statistics section”. All presented continuous data were confirmed to be parametric distributed with the Shapiro-Wilk-Test and are consequently presented as mean ± standard deviation. Statistical comparison between groups were performed with a student’s t-test (for parametric distributed data) or Man-Whitney-U test (for non-parametric distributed data).
We adapted the statistics section as follows:
Page 5, lines 1--9
“Continuous data are presented as mean ± standard deviation (SD) - their normality distribution was confirmed with a Shapiro-Wilk-Test. Categorical variables are displayed as numbers (percentage). Statistical comparison between independent groups was performed with the students t-test for parametric distributed data and with the Man-Whitney-U test for non-parametric distributed data. For the outcome analysis, Cox regression models were calculated. Time-to-event after first implantation was estimated by applying the Kaplan-Meier method. A p-value <0.05 was considered statistically significant. The statistical analyses were conducted with the STATA 17 software package (StataCorp LCC, Lakeway Drive, Texas, USA).”
- Comment 2: There are wrong values in the results section. Table 1: CRT was implanted in 26, but biventricular ICD system was implanted in 24. The total number of “indications for implantation” is 66 (24 + 14 + 26). These results must be checked and corrected. The same in Table 2 and throughout the text.
Answer: Thank you for pointing out this problem. In total, 14 patients underwent ICD implantation for secondary prevention and 50 patients for primary prevention. Overall, 24 patients (out of 64) had an additional indication for CRT.
We apologize for this error and changed the results section and tables as followed:
Page 6, lines 10-13
“ICD was implanted for secondary prevention in 14 patients (22%) and for primary prevention in 50 patients (78%). […] 24 patients were treated with cardiac resynchronization therapy.”
Table 1: Baseline-characteristics
|
|
n=64 |
|
Age (years) |
66.7 ± 8.7 |
|
Female |
19 (29.7) |
|
BMI (kg/m2) |
27.4 ± 5.8 |
|
LVEF (%) |
32.6 ± 10.5 |
|
History of inappropriate shocks |
4 (6.3) |
|
Underlying heart disease |
|
|
Ischemic |
34 (53.1) |
|
DCM |
26 (40.6) |
|
Other* |
4 (6.25) |
|
Indication for ICD |
|
|
Primary prevention |
50 (78.2) |
|
Secondary prevention |
14 (21.8) |
|
CRT |
24 (37.5) |
|
Implanted ICD system |
|
|
Single Chamber |
28 (43.8) |
|
Dual Chamber |
12 (18.8) |
|
Biventricular |
24 (37.5) |
BMI = Body mass index, LVEF = left ventricular ejection fraction, DCM = dilated cardiomyopathy, CRT = cardiac resynchronization therapy, ICD = intracardiac cardioverter defibrillator
* 2 patients with Brugada syndrome, 1 patient with hypertrophic cardiomyopathy and 1 patient post myocarditis
- Comment 3: The quality of Figure 2 is very low. Moreover, I think both Figures 1 and 2 are unnecessary in the manuscript and should be removed. Lines 64-65 and 74-75 should also be removed, consequently.
Answer: We removed figure 2 and the corresponding lines in the manuscript. Figure 1 may be of interest to some readers, who are not familiar with the Biotronik ICD leads, as it provides some interesting background information. We therefore included Figure 1 in the supplements and changed lines 74-75 in the manuscript as follows:
Page 4, lines 7-8
“ The Biotronik tachy-lead history and lead characteristics are displayed in supplementary figure 1.”
- Comment 4: Line 136 “at least form on” I suppose it should be “from”
Answer: Thank you for this hint, we changed the sentence according to your suggestion as follows:
Page 13, lines 13-14
“14 patients (26.6%) suffered at least from one episode of sustained ventricular tachycardia, which were successfully terminated with antitachycardia pacing (ATP) or a shock delivery. “
- Comment 5: Lines 138-139 and Figure 3: time-to-event curve (Kaplan-Meier) has no relation to the aim of the study.
Answer: We agree that the time-to-event curve (Figure 3) is not of main interest for our study. However, death rate is in general an interesting information and a marker of (preexisting) morbidity. Further, failure to provide ICD therapy may ultimately lead to death. Also, inappropriate shocks have been associated with increased mortality. Based on the above reasoning, we have left figure 3 in the manuscript, but reduced in size as suggested by Reviewer 2.
As figure 1 is now in the supplements and figure 2 was removed from the script, figure 3 is renamed to “figure 1”.
- Comment 6: Ref 18 is incomplete.
Answer: We apologize for the missing information and corrected reference 18 as follows:
Page 13, lines 44-46
“Seiler, T., et al., Recurrent implantable cardioverter-defibrillator shocks due to automatic deactivation of a right ventricular lead noise discrimination algorithm. HeartRhythm Case Rep, 2022. 8(10): p. 695-698.”
Round 2
Reviewer 1 Report
Authors have addressed most of my previous comments and improved the quality of the manuscript.
In line 66 (Abstract), however, there is still an incorrect number of patients - “26 patients treated with CRT” ; amend into 24.
After this correction, the manuscript will be ready for publication.
Author Response
We apologize for this error and changed the number of patients receiving CRT in the abstract from 26 to 24.
Reviewer 2 Report
I would like to thank the authors for their clear answers to all my comments. My remaining suggestion is to include some additional details from their replies in the manuscript. For example, the reply to comment 2 explaining the reason for excluding some events as markers of lead quality or longevity should be included in the manuscript.
Author Response
Thank you for this important advice. We added the following information in the methods section:
"Revisions due to lead dislodgements or perforations were not considered as failures, as they are mostly related to poor implantation quality and not directly related to longevity or quality of ICD leads."